# Crystal Structure Evolution, Microstructure Formation, and Properties of Mechanically Alloyed Ultrafine-Grained Ti-Zr-Nb Alloys at 36 ≤ Ti ≤ 70 (at. %)

**DOI:** 10.3390/ma13030587

**Published:** 2020-01-27

**Authors:** Mateusz Marczewski, Andrzej Miklaszewski, Xavier Maeder, Mieczyslaw Jurczyk

**Affiliations:** 1Institute of Materials Science and Engineering, Poznan University of Technology, Jana Pawla II No 24, 61-138 Poznan, Poland; andrzej.miklaszewski@put.poznan.pl (A.M.); mieczyslaw.jurczyk@put.poznan.pl (M.J.); 2Laboratory of Mechanics of Materials and Nanostructures, Empa, Swiss Federal Laboratories for Materials Science and Technology, Feuerwerkerstrasse 39, CH-3602 Thun, Switzerland; xavier.maeder@empa.ch

**Keywords:** metals and alloys, mechanical alloying, X-ray diffraction, phase transition, powder metallurgy

## Abstract

Titanium β-type alloys are preferred biomaterials for hard tissue replacements due to the low Young modulus and limitation of harmful aluminum and vanadium present in the commercially available Ti6Al4V alloy. The aim of this study was to develop a new ternary Ti-Zr-Nb system at 36 ≤ Ti ≤ 70 (at. %). The technical viability of preparing Ti-Zr-Nb alloys by high-energy ball-milling in a SPEX 8000 mill has been studied. These materials were prepared by the combination of mechanical alloying and powder metallurgy approach with cold powder compaction and sintering. Changes in the crystal structure as a function of the milling time were investigated using X-ray diffraction. Our study has shown that mechanical alloying supported by cold pressing and sintering at the temperature below α→β transus (600 °C) can be applied to synthesize single-phase, ultrafine-grained, bulk Ti(β)-type Ti30Zr17Nb, Ti23Zr25Nb, Ti30Zr26Nb, Ti22Zr34Nb, and Ti30Zr34Nb alloys. Alloys with lower content of Zr and Nb need higher sintering temperatures to have them fully recrystallized. The properties of developed materials are also engrossing in terms of their biomedical use with Young modulus significantly lower than that of pure titanium.

## 1. Introduction

Titanium appears in two different allotropic forms. At low temperatures, it has a closed packed hexagonal crystal structure (space group: P63/mmc–hcp), which is known as α, whereas above 882 ± 2 °C, it has a body-centered cubic structure (space group: Im-3m – bcc) termed β. The alloying elements, such as N, O, and Al, tend to stabilize the α phase, whereas elements V, Cr, Nb, and Mo stabilize the β phase [1,2,3,4]. Pure titanium and Ti-6Al-4V alloy are the main materials in the medical field. A component such as vanadium is described to be cytotoxic. The release of its ions to the human body system during implant wear can lead to neurodegenerative diseases such as Alzheimer’s disease. Furthermore, some of the researches also revealed the severe danger of the DNA damage caused by the Ti-6Al-4V alloy [5,6,7]. Vanadium ions were also proven to be toxic to the fibroblasts causing the significant limitation of their viability [8]; therefore, preparation of Ti alloys without these additives should be the prior goal of most researches. One of the possibilities is to produce β-type titanium alloys with low toxicity and high biocompatibility as Ti-Zr-Nb alloys. To stabilize Ti(β) it is necessary to form the alloys based on the following elements [9].

(a) Binary alloys: Ti-Nb, Ti-Mo, Ti-Ta, Ti-Zr, Ti-Mn, Ti–Cr.

(b) Ternary alloys: Ti-Nb-Mo, Ti-Nb-Pd, Ti-Nb-Zr, Ti-Nb-Sn, Ti-Nb-Ta, Ti-Nb-Fe, Ti-Mo-Zr, Ti-Mo-Nb, Ti-Cr-Al, Ti–Cr–Nb, Ti-Cr-Sn, Ti–Mn–Al, Ti-Ta-Nb, Ti-Ta-Sn, Ti-Ta-Zr, Ti–Mn–Fe, Ti–Sn–Cr.

(c) Quaternary alloys: Ti-Ta-Sn-Zr, Ti-Nb-Zr-Sn, Ti-Nb-Zr-Fe, Ti-Nb-Ta-Zr, Ti-Mo-Zr-Fe, Ti-Fe-Ta-Zr, Ti-Cr-Mn-Sn.

Two of the elements stabilizing Ti(β) are niobium and zirconium. Both have been proven to have higher biocompatibility than aluminum and vanadium. The biocompatibility of niobium was also concerned to be higher than titanium [10]. Ti-Zr [11,12,13,14,15] and Ti-Nb systems [16,17,18,19,20,21] were already examined in several articles. It was reported that Ti-Zr alloys can improve biocompatibility properties of pure titanium, their mechanical strength and grind ability [13,14]. Additionally, it can be observed that the content of Zr in the binary Ti-Zr alloys cannot provide Ti(β) formation at room temperature. Zr allows only to decrease the temperature of Ti(α)-Ti(β) phase transition to 806 °C [11,12]. On the other hand, Ti-Nb alloys allow improving the mechanical properties of titanium. Niobium increases the hardness of the alloy and decreases the compression modulus. Content of niobium influenced also compression strength and wear resistance [17]. Niobium forms also isomorphous phase with titanium providing good Ti(β) stabilizing effect and appearance of this phase at room temperature. Ti-Nb alloys can form double Ti(α) + Ti(β) phases and single Ti(β), Nb phases [21]. The matter of great importance is that crystal structure influences Young’s modulus which is lower with the addition of β-stabilizers. It was observed that Ti(β) alloys and near Ti(β) alloys can be characterized by the lower elastic modulus, even multiple times lower than commercially pure titanium (especially for alloys with niobium and zirconium content) [3,22]. It limits the possibility of mechanical unfit and loosening of the implant (so-called stress shielding effect) [1,2]. 

The Ti-Nb-Zr alloys need to be well treated after alloying to provide the correct structure. The properties of the β-type alloys are susceptible to heat and mechanical treatment. Ti-39Nb after repeated several times cold rolling and heat treatment had the single β structure with low Young modulus equals to 39 GPa. This is mainly because of the high density of the defects as dislocations and grain boundaries [23]. The properties of casted Ti–15Zr–5Cr–2Al alloys were also drastically changed after water quenching [24]. Titanium alloys in Ti-Nb-Zr system could be used as the coating. It is provided that this type of modification will improve the surface properties of the alloy. Moreover, depending on the type and the parameters of the process it is possible to control the properties and the microstructure of the manufactured implant layer [25]. To develop biomaterials properties the use of mechanical alloying (MA) should be considered [26]. Phase equilibria in the Ti-Nb-Zr system with the powders produced by this technology has not been investigated so far. However, these tests are relevant to determine the influence of the process parameters on the crystal structure of these materials. Recently, other β-type materials were tested as Ti-Fe-Zr alloys [27] and the properties of the Ti-Nb-Zr alloys were introduced only in two different chemical compositions: Ti14Zr16Nb and Ti23Zr25Nb [28]. Many β-type titanium alloys are still under investigation as selective laser melted metastable Ti–37Nb–6Sn [29], laser powder bed fused Ti-18Zr-14Nb [30] with nano α-phase precipitates, vacuum arc melted Ti-32Nb-(2, 4) Sn with nano α-phase precipitates [31], mechanically alloyed Ti-Mo alloys [32]. Low Young modulus alloys are not mentioned to be only the titanium-based alloys. Mechanical properties of materials as Zr-12Nb-4Sn seems to be also interesting [33]. 

Nowadays, the majority of Ti-Nb-Zr alloys are produced by traditional methods as vacuum arc remelting (VAR) [34,35]. Mechanical alloying allows us to form the nonequilibrium structures (nanocrystalline, ultrafine grain structure, etc.) influencing the improvement of some properties. The process can be controlled by the following parameters: time of milling, the ball to powder mass ratio, type of the mill, milling speed, etc [26,36]. Ti-Nb-Zr alloys have been produced by mechanical alloying but the influence of Nb and Zr on the crystal structure has not been examined [27]. The hardness of Ti13Zr20Nb (at. %) has been estimated as 660 HV, which is higher than the hardness of traditional alloys [37,38]. Recently Bai et al. investigated the diffusion behaviors of Zr and Nb in β-Ti alloys and developed an atomic mobility date base for the bcc phase in the Ti-Zr-Nb system using the LALPHAD method [39]. Additionally, the properties of ternary Ti-Nb-Zr alloys with porous structures synthesized by a magnetron sputtering method was studied [40]. It was shown that the porous structure was also dependent on the alloy composition. The Young’s modulus of ternary thin films was in the range of 80 to 95 GPa. 

The aim of the current study was the synthesis of ultrafine-grained Ti-Zr-Nb alloys at 36 ≤ Ti ≤ 70 (at. %) by mechanical alloying and powder metallurgy methods. Proposed in this work broad stabilizing elements composition was chosen to include both dual-phase and single-phase Ti-Zr-Nb alloys structure evolution as also its microstructure formation and properties comparison studies. The influence of Zr and Nb contents with a heat treatment temperature on phase transitions of Ti(α) to Ti(β), with a microstructure examination, were studied. Yet to the authors’ knowledge, there have been no papers regarding the addition of Zr and Nb to Ti-based alloys in very wide concentrations to have appeared until now.

## 2. Materials and Methods 

### 2.1. Chemicals and materials

The following powders were used to produce the materials: the commercial Ti (Alfa Aesar, Haverhill, MA, USA, 99.9% purity, CAS:7440-32-6), Nb (Sigma Aldrich, St. Louis, MO, USA, 99.8% purity, CAS:7446-03-1) powders and Zr fillings from a sponge (Sigma Aldrich, ≥ 99%, CAS:7440-67-7). The experiments were carried out on 9 different Ti-Zr-Nb type alloys (Table 1). The experiments were arranged at three stages: (1) Ti-Zr-Nb (Nb: 16-34 at. %; Zr: 14-30 at. %) powders preparation by mechanical alloying for 10 h, (2) sample consolidation by cold pressing and sintering at the temperature range of 600 to 1000 °C, and (3) materials characterization (phase structure analysis with X-ray diffraction, wettability, and surface free energy, corrosion resistance, hardness, and nanoindentation, EDS and EBSD).

### 2.2. Specimen Preparation

SPEX 8000 Mixer Mill (SPEX^®^ Sample Prep, Metuchen, NJ, USA) and round-bottom stainless vials were used for the mechanical alloying process in an argon atmosphere. Glove box (LabMaster 130) filled with argon atmosphere (O_2_< 2 ppm and H_2_O < 1 ppm) was used to weigh, blend, and pour (into vials) the elemental powders (Ti, Nb, and Zr). MA process was lasted up to 10 h in all cases. All the powders turned into bulk specimens of 6 mm diameter and 4 mm height. The pressure of 600 MPa was used for cold compaction to consolidate the powders. The specimens were next sealed in a quartz tube filled with an argon atmosphere. Specimens were sintered in five temperatures: 600, 750, 800, 850, and 1000 °C within a 30 min heating being sealed in a quartz tube filled with an argon atmosphere. Specimens were fast cooled in water.

### 2.3. Materials Characterization

Panalytical Empyrean XRD equipment with Cu Kα radiation (Almelo, Netherlands) was used to evaluate the structure of the specimens at room temperature during different processing stages. XRD measurements was done as follows; voltage 45 kV, anode current 40 mA, 2 theta range 30 – 80[°], time per step 60.325 [s/step], and step size 0.0334[°]. Background, profile coefficients, lattice parameters, linear absorption coefficients, and other variables were refined to obtain the spectra. Crystallite size and lattice strain of mechanically alloyed powders were estimated by the Williamson–Hall (W–H) analysis method [41] with the use of the uniform deformation model (UDM) [41] linearly fitting points assigned to the different diffraction lines. Rietveld analysis [42] was done to estimate the lattice parameters and phase quantity by Maud software. The following analyzed structural models were used in this approach: Ti(α) (ref. code 01-071-4632), Ti(β) ref. code 01-074-7075), Nb0.81Zr0.19 (ref. code 00-049-1455), Zr (ref. code 01-088-2329), and NbZr (ref. code 01-071-9970). The calculated pattern of the model structure was fitted by minimization of the sum of the squares and using Marquardt least-squares algorithm [43]. High goodness of fit (χ2<3) was achieved. Pattern fitting parameters: Rwp—weighted pattern residual indicator; Rexp—expected the residual indicator: S—the goodness of fit were revealed.

The chemical composition was examined by the scanning electron microscope (SEM, VEGA 5135, Tescan, Brno, Czech Republic) with the energy dispersive spectrometer (EDS, PTG Prison Avalon, Princeton Gamma Tech., Princeton, NY, USA) calibrated by a typical Cu calibration procedure. The parameters of the measurement are as follows; accelerating voltage: 20 kV; working distance: 23 mm; spot size: 160 nm. The microstructure of the alloys was characterized by electron backscatter diffraction (EBSD), using a Tescan Mira microscope (Brno, Czech Republic) and Digiview V camera from EDAX (Mahwah, NJ, USA). The crystal orientation maps were acquired using electron beam conditions of 20 kV and 10 nA, with 100 nm step size. The minimum misorientation angle for the grain calculation was 5° and with a minimum of 8 pixels, discarding the grain smaller than 300 nm in diameter. No cleaning procedure was applied to the maps. The contact angle (CA) of the surfaces was recorded by the optical system with a digital camera (Kruss-DSA25, Krüss, Hamburg, Germany) and measured by dedicated software (Kruss-Advanced 1.5, Krüss, Hamburg, Germany). The specimens were polished with Al_2_O_3_ suspension, flushed with alcohol, and dried before the measurement. Determination of contact angles for diiodomethane and glycerol was done with the ellipse fitting method [44] and 2 µL drop. Surface free energies were calculated based on the results for both fluids. The experiments were repeated three times for each specimen to determine the uncertainty and standard deviation for each measurement and all at ambient conditions (23 °C). For the specimen corrosion resistance analysis in the Ringer’s solution (NaCl: 9 g/L, KCl: 0.42 g/L, CaCl2: 0.48 g/L, NaHCO3: 0.2 g/L) at ambient conditions (23 °C). the potentiodynamic method was used. The experiments and analysis were conducted on Solartron 1285 potentiostat (Solartron Analytical, Farnborough, UK) with dedicated Corrware and Corrview software (Solartron Analytical, Farnborough, UK). Ag/AgCl electrode was used as a reference electrode in the measurements. The corrosion potential and current were calculated with Tafel curves. The specimens were polished with grinding paper 600 grit and clean with ethanol inside the ultrasonic bath before each measurement before starting. Potentiodynamic tests were preceded with open circuit potential measurement for 60 min. Every specimen was measured three times to estimate uncertainties and standard deviation.

Vickers microhardness (HV) of the sinters was measured using an Innovatest Nexus microhardness tester (Innovatest, Maastricht, Netherlands) with an applied load of 300 g and loading time 10 s. The modulus analysis of selected specimens based on the EIT (indentation modulus) estimation, was realized on Fischerscope HM2000 nanoindenter with Vickers tip located in Bern University of Applied Sciences (Biel, Switzerland). The measurement was carried out with nanoindenter using DIN 50 359/ISO 14577 standard, and load parameters of the test as: F (the max load with the loading time) = 300 mN/20s, C (creep time at the max load) = 5 s.

## 3. Results

Synthesis of ultrafine-grained Ti-Zr-Nb alloys by mechanical alloying and powder metallurgy method was the goal of this research, see Table 1. Crystal structure and its changes during milling were deeply examined. Diffractograms of the selected Ti-Zr-Nb alloy powders with the final single phase beta-structure (Ti23Zr25Nb) and dual-phase alpha + beta structure (Ti14Zr16Nb) milled for different times were presented in our previous article [28]. After 15 min of milling, the positions of the 2θ peaks of Ti-Zr-Nb mixtures are the same as that of the elemental powders, indicating that no significant reaction had occurred during milling. The hexagonal Zr (101) plane is barely visible after 3 h of milling. Moreover, 6 h of mechanical alloying leads to the formation of Ti(β) for all produced compositions. Zr content influences the angle position of the newly formed Ti(β) phase peaks moving them to the higher values. (Figure 1 and Figure 2). It can be seen that after 10 h of MA of Ti14Zr16Nb and Ti23Zr16Nb there are peaks of Ti(α) and Ti(β). Additionally, in the case of Ti30Zr17Nb, Ti14Zr25Nb, Ti23Zr25Nb, Ti30Zr26Nb, Ti13Zr33Nb, Ti22Zr34Nb, and Ti30Zr34Nb mixtures all three peaks emerge. These peaks mark to coincide with the peaks of Ti(β) phase suggesting that these mixtures could be of a bcc structure induced by deformation during the mechanical alloying. 

Mechanical alloying was described as a high energy milling process in which powder particles were subjected to repeated cold welding, fracturing, and rewilding [26]. The high plastic deformations of the powders result in a high density of dislocations and, subsequently, subgrains formation that may finally lead also to desired phase Ti(β) formation [45]. Our studies have shown that the mechanical alloying of mixtures of Ti, Zr, and Nb metal powders produces the crystal phases depending on the milling conditions and starting composition. No amorphous phases have also been observed in milled powder mixtures. Mechanical alloying allows inducing allotropic transformation of titanium (from hexagonal α to cubic β). Both the content of beta-stabilizers (Zr and Nb) and time help to control this transformation. Based on the Williamson–Hall model [41] after 10 h of milling crystallite size was estimated in the range of 14 to 28 nm. Moreover, the application of mechanical alloying increases the microstrains in the produced alloys (on average 9 – 23 × 10^−3^) – Figure 3. Proper choice of the process parameters and alloy composition allows producing crystalline Ti(β) phase. 

To produce bulk specimens of mechanically alloyed powder: cold pressing and sintering were conducted. Heating was done for 0.5 h and at five different temperatures: 600, 750, 800, 850 and 1000 °C. The process was conducted at the argon atmosphere to prevent the materials from oxidation. The diffractograms of alloys sintered at the same temperature of 750 °C were presented in Figure 4. The diffractogram relations of two selected specimens with single-phase beta structure (Ti23Zr25Nb) and dual-phase alpha+beta structure (Ti14Zr16Nb) with the sintering temperature were presented in our previous article [28]. Additionally, lattice constants and phase amounts relations based on a composition and processing parameters of the materials were presented in Table 2. After consolidation, two groups of produced materials can be distinguished:

(a) Ti14Zr16Nb, Ti23Zr16Nb, Ti14Zr25Nb, and Ti13Zr33Nb alloys with visible traces of Ti(α) (ref. code 01-071-4632), and Nb0.81Zr0.19 (ref. code 00-049-1455),

(b) Ti30Zr17Nb, Ti23Zr25Nb, Ti30Zr26Nb, Ti22Zr34Nb, and Ti30Zr34Nb alloys with visible traces of Zr (ref. code 01-088-2329), NbZr (ref. code 01-071-9970), and Nb0.81Zr0.19.

Single beta alloys were produced at some specific sintering conditions which confirm the high influence of consolidation parameters on the crystal structure of the materials. Ti23Zr25Nb and Ti30Zr26Nb alloys needed the sintering temperature in the range of 600 to 800 °C, Ti22Zr34Nb and Ti30Zr34Nb in the range of 600 to 750 °C and Ti30Zr17Nb—600 °C. In higher temperatures, pure Zr was present in the structure of alloys. It was observed in almost all materials except these with the smallest concentration of Zr: Ti14Zr16Nb, Ti14Zr25Nb, and Ti13Zr33Nb. This phase appears mainly in the temperatures above 850 °C. The increasing concentration of Ti(β) phase could be easily connected with the increase of the sintering temperature. All alloys, except Ti14Zr16Nb with the lowest content of beta-stabilizers, have the pseudo-beta structure with Ti(β) phase content above 97%.

The chemical compositions of alloys were confirmed with EDS. All the spectra and results were presented in Figure 5.

The microstructure of the sintered specimen was determined by EBSD (Figure 6). Both alloys show similar mean grain sizes in the order of 1.2 to 1.7 µm. The black areas in the maps are miss-indexed points due to too small grains which cannot be measured by classic EBSD. The grain smaller than 300 nm has been discarded from the grain size distribution shown in Figure 7. The dual-phase Ti14Zr16Nb alloy shows an alpha fraction of ~39%. The beta phase in this alloy contains a lot of low angle boundaries, as seen in the misorientation angle distribution in Figure 7. The alpha phase shows a misorientation angle distribution similar to the random one. The single beta phase Ti23Zr25Nb alloy shows some low angle boundaries in the larger grains, but to a less extent than the beta phase of the Ti14Zr16Nb alloy. 

Hardness and surface free energy by diiodomethane and glycerol contact angles of the Ti-Zr-Nb alloys were determined and the results listed in Table 3. Hardness for alloys was approximate 400 HV0.3. Contact angles for all alloys were much lower than 90° pointing their good wettability. Ti30Zr34Nb alloy was characterized by the highest value of surface free energy among all materials. The corrosion resistance of this alloy in the Ringer solution is also the highest which is indicated by the highest value of corrosion potential and the lowest value of corrosion current. All produced materials passivate in the higher potentials making their potentiodynamic curves not so different than that of pure titanium (Figure 8). This phenomenon is identified with no corrosion current increase in the anodic range of potentiodynamic curves and corresponds with the formation of the particular compounds on the material’s surface, preventing them from further corrosive wear.

Young modulus of all the β and pseudo-β Ti30Zr17Nb, Ti23Zr25Nb, Ti30Zr26Nb, and Ti30Zr34Nb is lower than 80 GPa and significantly lower than for dual-phase Ti14Zr16Nb alloy (Figure 9). There are no meaningful differences between the mechanically alloyed Ti-Nb-Zr alloys in terms of that property. Moreover, all the E modulus are nearly twice lower than pure titanium: 141 GPa [28].

## 4. Discussion

Presented results clearly demonstrate that the powder manufacturing route supported by the mechanical alloying process allows the production of β-Ti-based alloys. Moreover, it was possible with the formation of ultrafine-grained structure. It could lead to better mechanical properties of produced alloys cause of the strengthening of the material with the use of the grain refinement mechanism.

In one of the latest papers of our group [28], the MA technique was used to manufacture Ti14Zr16Nb and Ti23Zr25Nb (at. %) alloys. A positive impact of using MA process on properties was due to the reduction of particle size, creation of new clean surfaces, and creation of various types of defects. These materials were prepared by the combination of mechanical alloying and powder metallurgy approach with cold powder compaction and sintering or interchangeably hot pressing. The mechanical alloying of Ti, Zr, and Nb for 10 h with additional hot pressing (71 MPa) and sintering for 10 min at 600 °C results in ultrafine-grained structure formation. It has been observed in the cited paper that the mechanically alloyed Ti23Zr25Nb material upon sintering at 600 °C for 10 min led to the formation of a single β-type phase alloy. Note that to control the crystal structure of Ti(β) alloys, processing parameters are a significant factor. These alloys are not only sensitive to the beta-stabilizers content but also to the sintering temperatures. However, Nb has much higher beta-stabilizing properties than Zr, which mainly leads to the spontaneous passivation of its alloys and does not form Ti(β) structure in the Ti-Zr binary alloys as a neutral element [46]. Henriques et al. synthesized the Ti–13Nb–13Zr alloy by conventional sintering [47]. The densification between 93 and 97% was achieved at a high sintering temperature of 1400 °C. Recently, nanostructured near-β Ti–20Nb–13Zr at. % alloy with nontoxic elements and enhanced mechanical properties have been synthesized by spark plasma sintering (SPS) of nanocrystalline powders obtained by mechanical alloying [37]. A nearly full density structure was obtained after SPS at 1200 °C. The microstructure of the obtained alloy is a duplex structure with the α-Ti (hcp) region having an average size of 70–140 nm, surrounding the β-Ti (bcc) matrix. The obtained alloy was chemically homogenized with a microhardness value (HV) of 660. Compared to these results, it is clear that we were able to obtain by the powder metallurgy approach nearly full density in the Ti–Zr-Nb alloys, at a much lower temperature (600 °C) [28]. The synthesized Ti23Zr25Nb alloy was chemically homogenized with a hardness value, HV0.3 of 407. The Ti, Zr and Nb are nontoxic elements [10]. Additionally, Nb in Ti–Zr-Nb system act as β-stabilizer and Zr acts as a neutral element for forming a homogeneous solid solution in the α- and β-phases [48]. On the other hand, Nb is found to reduce the elastic modulus when alloyed with titanium [49]. Studies on Ti–Zr-Nb systems for medical applications have shown that α↔β phase transformations are sensitive not only by chemical composition but also by heat treatment parameters and cooling rates. 

The Zr content in the Ti-Nb-Zr alloys not only allows the formation of the single-phase β-type alloys, but also influences the lattice constant of both Ti(α) and Ti(β)-phase. The increase of the Zr clearly expands the volume of the cell (Table 2). Nb in Ti-Nb-Zr does not have any impact on the Ti(β) lattice constants but increases the amount of the Ti(β) phase and volume of Ti(α) phase (in smaller range than Zr). The Ti-Zr alloys allow only to form a single-phase α-type alloys without β-phase presence because of no stabilization effect of Zr without the present of Nb in the alloy composition could not be observed. The volume of Ti(β) phase cell in the produced alloys varies from 36.65(1) Å^3^ to 38.51(1) Å^3^ and volume of Ti(α) phase vary from 36.73(2) Å^3^ to 37.89(26) Å^3^. The structure of produced alloys depends also on the temperature of sintering. It was observed that in the produced single β-phase alloys beneath 25 at. % Nb content (Ti23Zr16Nb, Ti30Zr17Nb, Ti23Zr25Nb Ti30Zr25Nb) in the temperatures above 750 –800°C lattice constant of Ti(β) phase decreases. In the dual-phase alloys (Ti14Zr16Nb, Ti14Zr25Nb) in the temperatures beneath 1000 °C the same situations always happen with the Ti(α) phase volume. The volume of Ti(β) phase grows in these two alloys continuously with the sintering temperature increase. In the alloys with Nb content from 23 at. % to 34 at. % Zr-phase appears in the higher temperatures. Moreover, in Ti22Zr34Nb alloy in the temperature of 600 °C the NbZr-phase diffraction peaks are visible and show the same group symmetry as Ti(β) phase but with a higher volume cell equals to 41.33(7) Å^3^. It is the transitional phase that appears because of limited solubility in the lower temperature of Zr and Nb in Ti solid solution. It disappears in higher temperatures forming a single β-phase structure in the temperature of 750 °C and 800°C.

The EBSD results show significant differences in microstructures between dual-phase Ti14Zr16Nb and a single β-phase Ti23Zr25Nb alloys. The misorientation distribution histograms can be easily contrasted (Figure 7). The recrystallization of Ti-Zr-Nb powders is driven by the energy stored in the mechanically alloyed materials coming from multiple crystal defects in the dominant form of the dislocations coming from the plastic deformation of powders during milling. These types of defects are considered to be the main part of the stored energy [50]. These alloys should be considered as high stacking fault energy materials with dynamic recovery and continuous dynamic recrystallization (CDRX) [50,51]. CDRX is the mechanism of low angle grain boundaries (LAGB) migration to the high angle grain boundaries (HAGB). It is clearly visible that CDRX in Ti14Zr16Nb alloy has not been completed cause a high amount of HAGB in comparison to the random distribution histograms. However, it was finished in Ti23Zr25Nb alloy with higher β-stabilizing Nb and Zr content. This type of histogram for this alloy matches the random one. The above result confirms that Ti14Zr16Nb alloys were not fully recrystallized in the provide sintering temperature of 750 °C because LAGB was not fully annihilated by HAGB. The thermal energy provided during sintering was not enough. However, the same temperature was enough to recrystallized and annihilate the HAGB in the Ti23Zr25Nb alloy. The Rietveld refinement results for Ti14Zr16Nb alloy (Table 2) shows the trend of the clear increase of Ti(β) phase fraction with sintering temperature. It is in parallel to the further recrystallization of this material. Nb and Zr limit the thermal energy provided to that materials during the sintering process and needed to fully recrystallize highly-defected Ti-Zr-Nb alloys after the mechanical alloying process which can be easily proven by the limitation of α→β phase transformation temperature in the binary phase Ti-Nb and Ti-Zr diagrams. That is highly dependent on their β-stabilizing properties. EBSD results prove also an obtained ultrafine grain structure of prepared by mechanical alloying, cold compaction and sintering approach materials, with grain size significantly lower than one micron (Figure 7).

Moreover, the produced β and pseudo-β alloy—Ti30Zr17Nb, Ti23Zr25Nb, Ti30Zr26Nb, and Ti30Zr34Nb—have interesting properties in terms of their biomedical applications. Their corrosion properties are nearly similar to that of the pure titanium which means that they are not going to behave much differently inside the human body. However, the most important remains a significant reduction in their Young modulus comparing to pure titanium. These properties limit the stress shielding effect and increase the osteointegration which should be also stimulated by the porosities in these materials [52]. The obtained results are between 67.9 (Ti30Zr34Nb) and 84.6 (Ti14Zr16Nb) which makes their modulus near about twice lower than that of commercially pure titanium. The modulus of β and pseudo-β alloys are also lower than that of dual-phase α+β Ti14Zr16Nb alloy, which proves clearly the influence of the crystal structure of these materials on their mechanical properties.

In this work, the mechanical alloying process followed by the pressing and sintering was applied for the preparation of the bulk Ti-Zr-Nb near-β type alloys. The developed ultrafine-grained Ti-based alloys can be used as the implant materials in dental and orthopedics applications. The special interest should be focused on modifying these alloys by the formation of biocomposite materials. Biocomposite formed by adding 45S5 Bioglass to these alloys during milling can lead to interesting results as higher biocompatibility, hardness, and lower Young modulus. The structure, microstructure, porosity, surface morphology, corrosion resistance, wettability, and in vitro cytocompatibility will be investigated and the results will be published independently.

## 5. Conclusions

The Ti-Zr-Nb near-β type alloy 36 ≤ Ti ≤ 70 (at. %) was synthesized by mechanical alloying and powder metallurgy method. The influence of Nb and Zr contents as also heat treatment on phase transitions (α→β) were studied. The main conclusions can be withdrawn: For the analyzed compositions, the Ti(β) –phase content can be increased with the longest processing time of mechanical alloyin; the formation of Ti(β) type alloys is possible after recrystallization of MA powders during its sintering in the wide range of process temperatures; with the increase of Zr and Nb amount in the Ti-Zr-Nb system, the Ti(β) phase content is higher and Zr has the main impact on the Ti(β) lattice constant and Nb on the beta-stabilizing effect; single-phase Ti(β) type Ti30Zr17Nb, Ti23Zr25Nb, Ti30Zr26Nb, Ti22Zr34Nb, and Ti30Zr34Nb alloys are synthesized after sintering at a low temperature (600 °C) for 30 min; the mechanical alloying and powder metallurgy approach remains beneficial in grain growth control and synthesis of ultrafine-grained Ti-Zr-Nb alloys for implant applications, the recrystallization process during sintering and final microstructure of Ti-Zr-Nb alloys clearly depends on the β-stabilizers content; and the developed ultrafine-grained Ti-Zr-Nb alloys could be used as the implant materials in dental and orthopedics applications because of their good properties as significantly reduced Young modulus in contrast to the commercially pure titanium and limitation of toxic elements as β-stabilizers.

## Figures and Tables

**Figure 1 materials-13-00587-f001:**
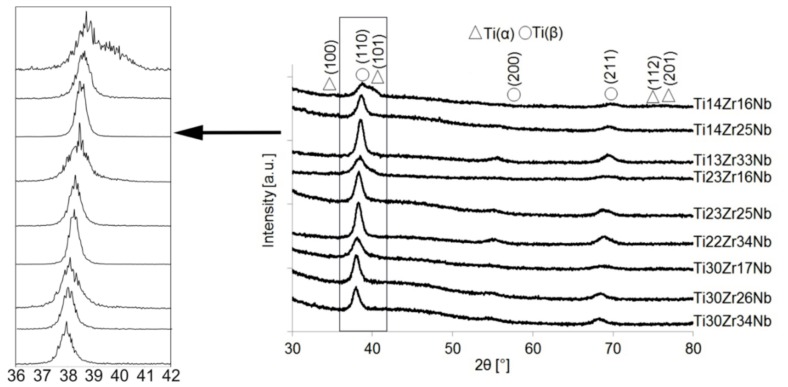
X-ray diffraction (XRD) spectra of Ti-Zr-Nb powders mechanically alloyed for 10 h.

**Figure 2 materials-13-00587-f002:**
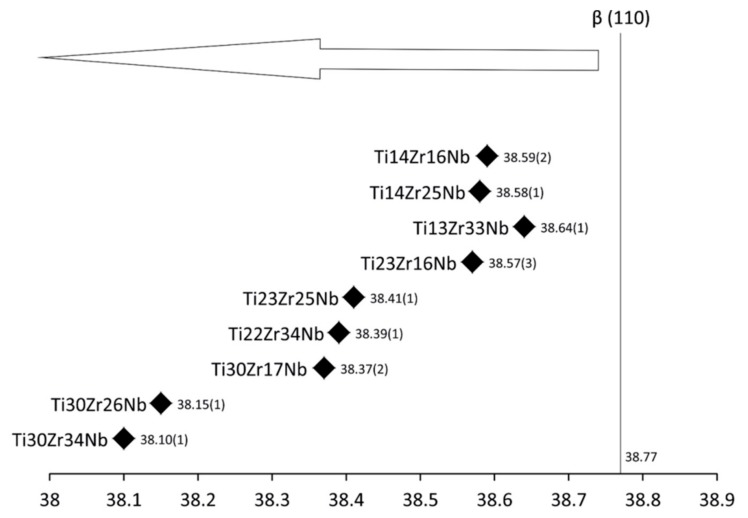
New phase Ti(β) 2 theta locations of studied alloys after 6 h of MA.

**Figure 3 materials-13-00587-f003:**
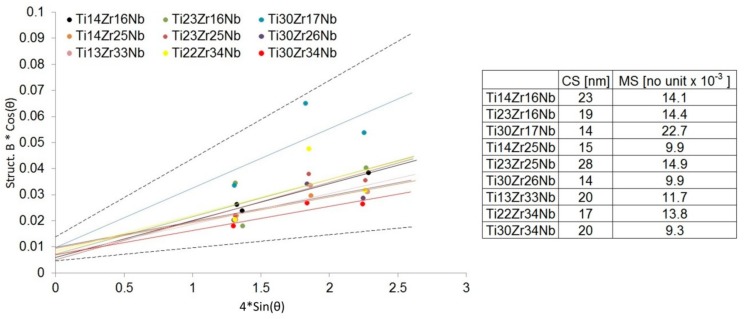
Linear Williamson–Hall plots with estimated crystallite size (CS) and microstrain (MS) factors based on the XRD spectra of Ti-Zr-Nb powder materials after 10 h of mechanical alloying.

**Figure 4 materials-13-00587-f004:**
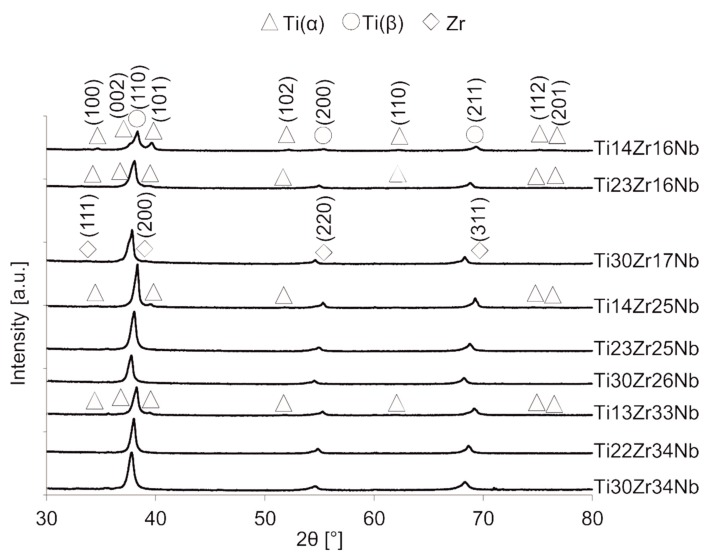
XRD spectra of bulk Ti-Zr-Nb specimens sintered at 750 °C.

**Figure 5 materials-13-00587-f005:**
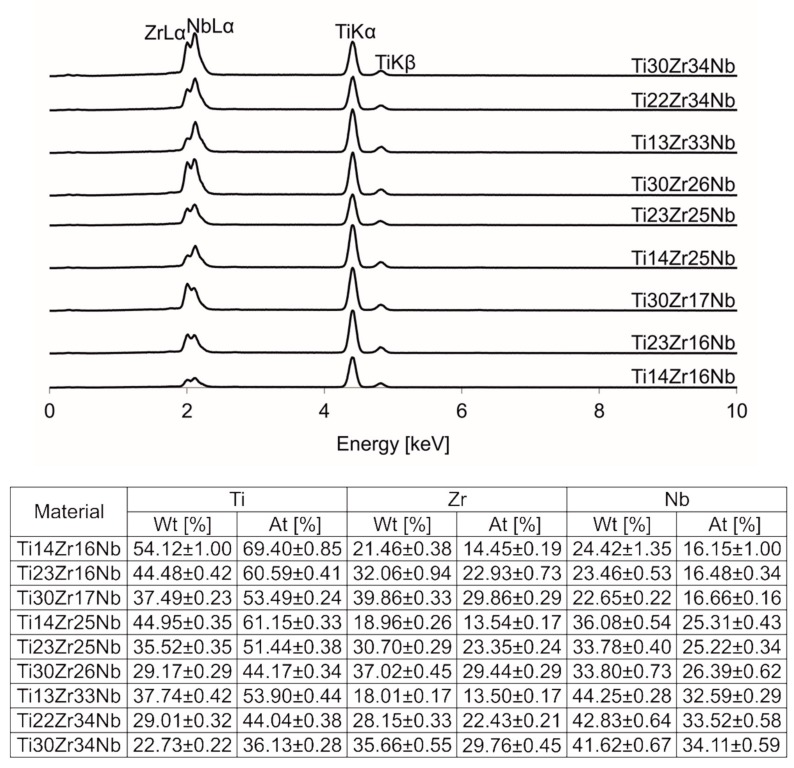
Energy-dispersive spectroscopy (EDS) spectra of Ti-Zr-Nb alloys.

**Figure 6 materials-13-00587-f006:**
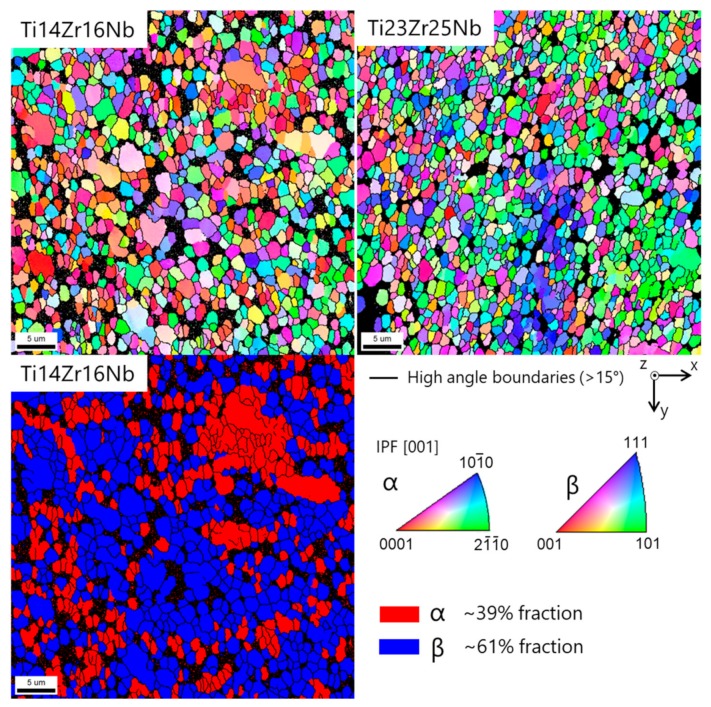
Top: Electron backscatter diffraction (EBSD) crystal orientation maps of α and β phases in Ti14Zr16Nb and Ti23Zr25Nb (only β phase) cold pressed and sintered at 750 °C for 30 min. The IPF color is displayed for the z-direction (cold pressing axis). Bottom: α and β phase distribution map in Ti14Zr16Nb.

**Figure 7 materials-13-00587-f007:**
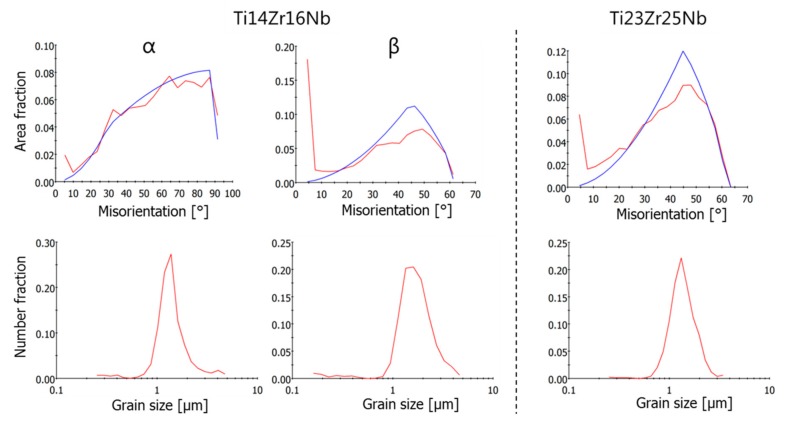
Misorientation angle and grain size distributions from EBSD measurements of α and β phases in Ti14Zr16Nb and Ti23Zr25Nb (only β phase) cold pressed and sintered at 750 °C for 30 min.

**Figure 8 materials-13-00587-f008:**
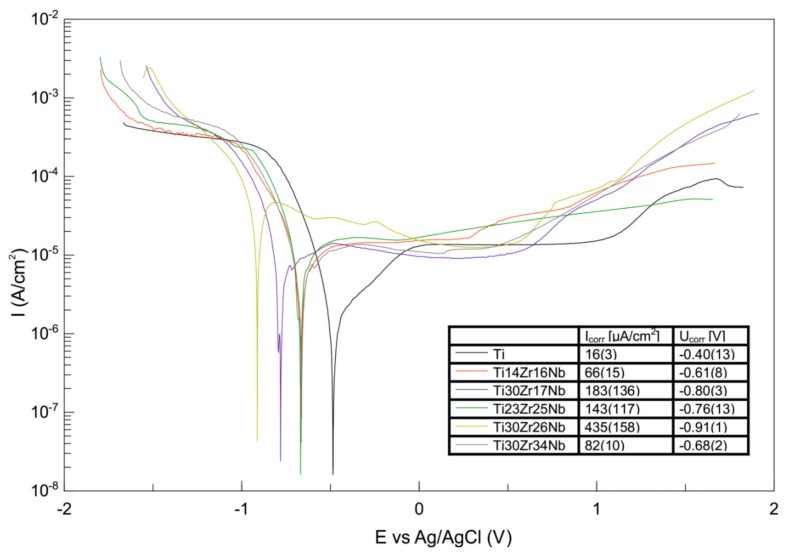
Potentiodynamic curves of bulk β and pseudo-β alloys Ti30Zr17Nb, Ti23Zr25Nb, Ti30Zr26Nb, Ti30Zr34Nb cold pressed and sintered at 750 °C for 30 min in comparison with dual-phase Ti14Zr16Nb processed in the same conditions. The results were obtained from the group of three replicated measurements.

**Figure 9 materials-13-00587-f009:**
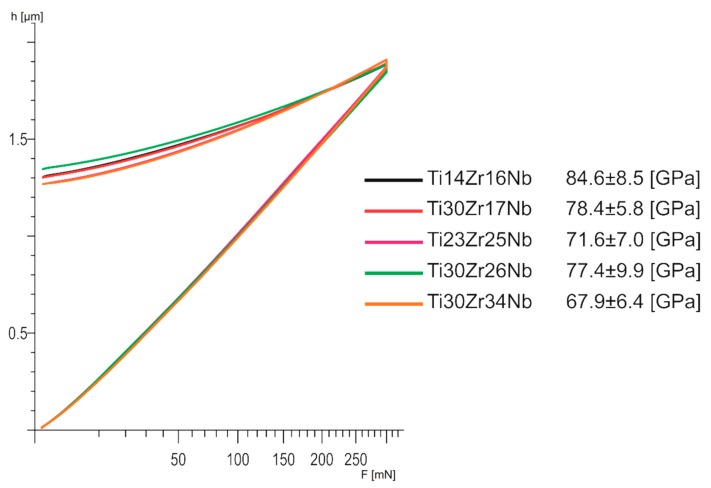
Load–depth curves of bulk β and pseudo-β alloys Ti30Zr17Nb, Ti23Zr25Nb, Ti30Zr26Nb, and Ti30Zr34Nb cold pressed and sintered at 750 °C for 30 min, in comparison with dual-phase Ti14Zr16Nb processed in the same conditions.

**Table 1 materials-13-00587-t001:** Chemical compositions of output powders needed to get following Ti-Nb-Zr alloys after all stages of specimen preparation by the application of SPEX 8000 Mixer Mill (total weight of milling powders: 4.5 g; ball-to-powder mass ratio: 7.5:1; milling time: 10 h).

Alloy	Ti	Zr	Nb
	wt %	wt %	wt %
Ti14Zr16Nb (at. %)	66	17	17
Ti23Zr16Nb (at. %)	57	26	17
Ti30Zr17Nb (at. %)	50	33	17
Ti14Zr25Nb (at. %)	57	17	26
Ti23Zr25Nb (at. %)	48	26	26
Ti30Zr26Nb (at. %)	41	33	26
Ti13Zr33Nb (at. %)	50	17	33
Ti22Zr34Nb (at. %)	41	26	33
Ti30Zr34Nb (at. %)	34	33	33

**Table 2 materials-13-00587-t002:** Crystallographic data analysis of bulk Ti-Zr-Nb specimens sintered at different conditions.

Specimen	PT [°C]	Ti(α)				Ti(β)			Additional Phase			R_wp_ [%]	R_exp_ [%]	S
									Nb0.81Zr0.19					
		a [Å]	c [Å]	V [Å^3^]	PA [%]	a [Å]	V [Å^3^]	PA [%]	a [Å]	V [Å^3^]	PA[%]			
Ti14Zr16Nb	600	2.9862(4)	4.7566(14)	36.73(2)	44.83	3.3218(3)	36.65(1)	55.17	-	-	-	4.86	3.35	1.45
	750	2.9853(4)	4.7761(15)	36.86(3)	26.54	3.3255(3)	36.78(1)	73.46	-	-	-	5.67	3.20	1.77
	800	2.9908(5)	4.7843(19)	37.06(3)	26.20	3.3313(3)	36.97(1)	73.80	-	-	-	6.28	3.65	1.72
	850	2.9903(11)	4.7817(41)	37.03(6)	9.94	3.3327(2)	37.02(1)	88.64	4.3477(25)	82.18(14)	1.42	7.35	3.39	2.17
	1000	2.9972(16)	4.7798(63)	37.18(9)	2.69	3.3357(2)	37.12(1)	96.02	4.3214(14)	80.70(8)	1.29	6.58	3.04	2.17
									**Zr**					
Ti23Zr16Nb	600	2.9966(8)	4.7892(33)	37.24(5)	24.90	3.3494(2)	37.57(1)	75.10	-	-	-	5.47	2.99	1.83
	750	2.9982(34)	4.8054(156)	37.41(21)	7.87	3.3510(2)	37.63(1)	92.13	-	-	-	6.58	3.03	2.17
	800	3.0246(41)	4.7830(196)	37.89(26)	9.65	3.3527(2)	37.68(1)	90.35	-	-	-	6.42	2.91	2.21
	850	-	-	-	-	3.3509(2)	37.63(1)	99.19	4.6102(17)	97.98(11)	0.81	5.01	2.95	1.70
	1000	-	-	-	-	3.3532(2)	37.70(1)	98.98	4.6051(14)	97.66(9)	1.02	6.33	2.89	2.19
									**Zr**					
Ti30Zr17Nb	600	-	-	-	-	3.3610(2)	37.97(1)	100	-	-	-	6.74	2.72	2.48
	750	-	-	-	-	3.3699(2)	38.27(1)	99.66	4.6157(29)	98.33(18)	0.34	6.73	2.84	2.37
	800	-	-	-	-	3.3688(2)	38.23(1)	98.71	4.6050(23)	97.66(15)	1.29	6.89	2.89	2.39
	850	-	-	-	-	3.3628(2)	38.03(1)	99.53	4.6054(22)	97.68(14)	0.47	6.62	2.77	2.39
	1000	-	-	-	-	3.3671(2)	38.17(1)	99.27	4.6092(13)	97.92(9)	0.73	7.79	2.65	2.94
									**Zr**					
Ti14Zr25Nb	600	2.9826(7)	4.7679(27)	36.73(4)	14.05	3.3232(2)	36.70(1)	85.95	-	-	-	5.50	2.99	1.84
	750	2.9949(13)	4.7679(55)	37.04(8)	6.98	3.3261(2)	36.79(1)	93.02	-	-	-	6.55	2.82	2.32
	800	2.9993(15)	4.7671(65)	37.14(9)	7.24	3.3280(2)	36.86(1)	92.76	-	-	-	6.35	2.97	2.14
	850	2.9992(22)	4.7653(87)	37.12(12)	4.68	3.3278(2)	36.85(1)	95.32	-	-	-	7.35	2.95	2.49
	1000	3.0091(43)	4.7939(183)	37.59(25)	3.23	3.3359(2)	37.12(1)	96.77	-	-	-	6.21	3.05	2.03
									**Zr**					
Ti23Zr25Nb	600	-	-	-	-	3.3440(1)	37.39(1)	100	-	-	-	4.86	2.67	1.82
	750	-	-	-	-	3.3495(2)	37.58(1)	100	-	-	-	5.19	2.60	2.00
	800	-	-	-	-	3.3524(2)	37.68(1)	100	-	-	-	5.05	3.08	1.64
	850	-	-	-	-	3.3500(2)	37.60(1)	99.50	4.6035(30)	97.56(19)	0.50	5.46	2.97	1.84
	1000	-	-	-	-	3.3500(2)	37.60(1)	99.07	4.6077(25)	97.83(16)	0.93	6.76	2.74	2.46
									**Zr**					
Ti30Zr26Nb	600	-	-	-	-	3.3694(2)	38.25(1)	100	-	-	-	5.61	3.23	1.74
	750	-	-	-	-	3.3720(2)	38.34(1)	100	-	-	-	5.36	3.02	1.77
	800	-	-	-	-	3.3753(2)	38.45(1)	100	-	-	-	6.20	2.62	2.37
	850	-	-	-	-	3.3720(2)	38.34(1)	99.35	4.6199(16)	98.60(10)	0.65	5.82	2.93	1.99
	1000	-	-	-	-	3.3770(2)	38.51(1)	98.66	4.6188(12)	98.53(8)	1.34	6.53	2.64	2.48
									**Nb0.81Zr0.19**					
Ti13Zr33Nb	600	2.9880(15)	4.7643(68)	36.84(9)	6.88	3.3242(2)	36.73(1)	93.12	-	-	-	5.09	3.04	1.67
	750	3.0022(14)	4.7679(56)	37.22(8)	5.60	3.3314(2)	36.97(1)	94.40	-	-	-	5.38	2.95	1.82
	800	3.0033(22)	4.7591(98)	37.18(13)	4.82	3.3271(2)	36.83(1)	95.18	-	-	-	5.82	2.76	2.11
	850	3.0103(16)	4.7667(73)	37.41(10)	6.94	3.3318(2)	36.99(1)	93.06	-	-	-	6.18	2.79	2.22
	1000	-	-	-	-	3.3327(2)	37.02(1)	99.44	4.3283(48)	81.09(27)	0.56	7.18	2.97	2.41
									**NbZr**					
Ti22Zr34Nb	600	-	-	-	-	3.3490(2)	37.56(1)	97.14	3.4573(18)	41.33(7)	2.86	4.66	2.73	1.71
									**Zr**					
	750	-	-	-	-	3.3512(1)	37.63(1)	100	-	-	-	4.93	2.98	1.65
	800	-	-	-	-	3.3515(1)	37.65(1)	100	-	-	-	6.13	2.64	2.32
	850	-	-	-	-	3.3527(2)	37.69(1)	99.78	4.5920(31)	96.83(20)	0.22	6.33	2.68	2.36
	1000	-	-	-	-	3.3537(2)	37.72(1)	99.63	4.5992(23)	97.29(15)	0.37	6.99	3.07	2.28
									**Zr**					
Ti30Zr34Nb	600	-	-	-	-	3.3528(2)	37.69(1)	100	-	-	-	5.03	2.44	2.06
	750	-	-	-	-	3.3677(2)	38.20(1)	100	-	-	-	4.6	2.56	1.80
	800	-	-	-	-	3.3698(1)	38.27(1)	99.15	4.6383(29)	99.79(19)	0.85	4.46	2.94	1.52
	850	-	-	-	-	3.3706(1)	38.29(1)	99.04	4.6236(14)	98.84(9)	0.96	4.33	2.71	1.60
	1000	-	-	-	-	3.3742(2)	38.42(1)	98.86	4.6288(8)	99.18(6)	1.14	4.37	3.21	1.36

**Table 3 materials-13-00587-t003:** Vickers hardness (HV0.3), surface free energy, diiodomethane, and glycerol contact angles for β and pseudo-β alloys Ti30Zr17Nb, Ti23Zr25Nb, Ti30Zr26Nb, and Ti30Zr34Nb, cold pressed and sintered at 750 °C for 30 min in comparison with dual-phase Ti14Zr16Nb processed in the same conditions. The results were obtained from the group of three replicated measurements.

Specimen	HV0.3	CA[M] Diiodomethane [°]	CA[M] Glycerol [°]	SFE [mN/m]
Ti14Zr16Nb	409 ± 17	53.4 ± 2.0	69.2 ± 2.2	35.0 ± 2.0
Ti30Zr17Nb	387 ± 14	59.8 ± 1.6	93.3 ± 3.3	29.1 ± 1.2
Ti23Zr25Nb	384 ± 13	54.0 ± 7.8	74.2 ± 6.8	34.4 ± 6.6
Ti30Zr26Nb	386 ± 15	65.1 ± 3.7	70.6 ± 2.5	30.8 ± 4.1
Ti30Zr34Nb	412 ± 17	48.0 ± 1.8	61.5 ± 6.2	40.0 ± 3.6

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
