# Peer review of "Crystal Structure Evolution, Microstructure Formation, and Properties of Mechanically Alloyed Ultrafine-Grained Ti-Zr-Nb Alloys at 36≤Ti≤70 (at. %)"

_materials, 2020, doi:10.3390/ma13030587_

Round 1
Reviewer 1 Report
The topic of the manuscript “
Crystal structure evolution, microstructure formation and properties of mechanically alloyed ultrafine-grained Ti-Zr-Nb alloys at 36≤Ti≤70 (at %) 4 for biomedical applications” is seems to be interesting and important. According to my opinion this study is a valuable work, the paper is well written, the research is well designed and decisions are justified, (I have some doubts about the paper which I listed below), therefore I suggest this paper for “major revision”:
I do not like the title, the authors did not mention the biomedical application in the manuscript, so this part could be strengthen or delete. The authors stated in the abstract that: “The results of experiments allow confirming the high dependence of crystal structure, microstructure and recrystallization process on the composition and sintering parameters of Ti-Zr-Nb alloys.” Of course it is the truth in the case of each alloys, so please avoid such a too general statements. The writing style of the authors are too “segmented”, please do not start a new paragraph after each sentence! Not the “articles” examine the alloys, but the researchers, please avoid such a mismatching frames. I do not feel the background of the scientific problem well presented. The authors discuss deeply the different Ti alloys and their applications, however about the current problems, question they gave only some lines. Please strengthen this part! 1 could be omitted, only write the sequence of the study the meaningless pics are not necessary. Please do not start new lines after the structural models. In Fig. 2 and 6, 7 I see 9 very small (almost unreadable) and almost the same graphs. Please show only the interesting parts, we will believe that you really measured them. Fig. 3 is a nice and meaningful fig. I do not like the style of the Conclusions section, it would be better to write it fluently, not such a listing. You could write something about the possible application of your results.
Reviewer 2 Report
Although the paper is interesting and provides valuable information it cannot be published before general re-editing.
The paper contains 12 figures but most of the figures are compound type figures and in each one of them exist 6 or more sub-figures. This makes the total number of figures unacceptably large for a scientific paper and makes the paper very difficult to be followed. Fig. 2 contains 9 subfigures, Fig.6 contains 9 sub-figures, Fig. 7-5 sub-figures, Fig.10-6, Fig.12-5!
In total there are 44 figures!
Many figures are presented as raw data and the important information from the figures is not extracted and post-processed. This makes current submission to look like a technical report but not as a scientific paper.
The same is valid for the data in the tables.
The authors must reduce stronly the number of figures to a reasonable number by which they can convey their message to the readers in a clear way. If they still want to show their raw results they can make an appendix with the raw experimental data but they cannot stay like that in the paper. The data should be processed, quantified and analyzed in debt and I cannot see this in this submission.
The parameters of the EDX analysis were not given in the description of the experimental part.
The parameters of the EBSD analysis are only partially given. Is there any data post-processing (clean up) and if yes, what kind of? What is the grain definition used to determine the grain size? There is no sample coordinate system shown on the inverse pole figure map although it was shown that the IPF map is <001> i.e. normal direction. How it was linked to the sample coordinate system? The average calculated grain diameter was claimed to be between0.8 and 1µm but from Figure 10 it seems to be larger than 1µm, more between 1.5 and 2µm. It is good authors to double-check the data.
Line 377-378: If the authors cannot measure the very small grains via EBSD because they are in the submicron range why they do not use a classical SEM? The grains with a size below 300nm should be visible in an appropriate imaging mode.
Line 407-409: the sentence requires language correction.
Reviewer 3 Report
Abstract. (1) At line 16, insert a phrase to explain concisely why beta-titanium alloys are preferred biomaterials for hard tissue replacements. (2) At line 28, alternative, more scientific wording is recommended in place of “really interesting”. (The later use of “really” in the text should also be avoided.)
Introduction. (1) At line 37, “etc” should be deleted at the two places; “such as” is sufficient when there are no more items to list. (2) In the second paragraph, there should also be comments about the reported health hazard concerns regarding the Al component in the widely used Ti-6Al-4V biomedical alloy. (3) At lines 50 and 51, there should be a reference for the important sentence stating that niobium and zirconium have been proven to have higher biocompatibility than aluminum and vanadium. (4) At line 56, use “cannot”. Contractions should be avoided in scientific writing. (There are other places in the manuscript where this change is needed.) (5) At line 72, the relevant specific “defects” in the crystal structure should be stated. (7) At line 101, should “bbc” be “bcc”? (8) The sentence at line 106 should be followed by a new explanatory sentence or phrase to explain why the alloy composition range from 36 – 70 at. % Ti was selected for study.
Materials and Methods. (1) At line 121, “BPR” used in Table 1 should be parenthetically defined here or in the text. (2) This reviewer prefers the use of “test specimen” or “specimen” rather than the intuitive term “sample”, since the latter has the meaning in statistics of a group of nominally identical, replicate, individual test specimens. (3) At lines 141 – 143, 150 and 166, it would be worthwhile for many readers if references were provided for the Williamson-Hall analysis method, the uniform deformation model, the Rietveld analysis method, the Marquardt least-squares algorithm, and the ellipse fitting method. (4) At lines 145 – 148, do the reference codes refer to the ICDD powder standards or to another set of powder diffraction standards? (6) At line 185, the nanoindentation testing parameters “F” and “C” should be defined.
Results. (1) At line 212, “ordered phase” presumably does not refer to a superlattice structure. Alternative wording should be used. (2) Do the different groups of X-ray diffraction patterns provide evidence of preferred crystallographic orientation? (3) For the sentence at lines 209 – 211, a reference should be provided. (4) At line 217, there should be a reference for the Williamson-Hall model. (4) For the table presented in Figure 8, does the accuracy of the EDS analyses, using the calibration procedure with the copper standard, warrant presenting the elemental concentrations to the nearest 0.01 percent? (5) At line 281, the description of “transparently passivating” should be clarified. (6) The number of replicate specimens or replicate measurements should be included with the information for Table 3 and Figure 11.
Discussion. (1) At line 330, references would be worthwhile to support the important sentence: “The Ti, Zr and Nb are non-toxic elements.” (2) At line 359, it would be worthwhile to specify the “multiple crystal defects”. (3) At lines 360 and 361, does Reference 43 state that these titanium alloys have high values of stacking fault energy? If not, this important point about the SFE should be referenced. (4) The authors should add a concluding paragraph that emphasizes the potential importance of their novel processing protocol for obtaining high-quality Ti-Zr-Nb biomedical alloys. For example, there are a variety of concerns with the casting of these alloys, using a dental casting machine that is designed for titanium and its alloys. This paragraph should also summarize the limitations of the study and provide suggestions for future research.
Round 2
Reviewer 1 Report
The authors answered my questions and corrected the paper, it is acceptable in the present form.
Author Response
Dear Reviewer 1,
Thank you very much once again for your previous comments considering our work. We really appreciate its acceptance.
Yours faithfully,
Mateusz Marczewski
Reviewer 2 Report
The authors addressed the reviewers comments almost completely and correctly.
However, a minor comment on their answer, regarding the grain size definition. Although the answer of the authors is absolutely correct , I afraid they did not understand that question. In EBSD characterization the grain size is defined by clearly mentioning the minimum number of pixels that can belong to a grain and by the minimum misorientation between the neighboring pixels inside the grain. Hence, the grain definition should contain these two parameters, namely “Minimum pixels per grain” and “min misorientation angle”. By default in EDAX- TSL software the min misorientation angle for a grain definition is 5° and the minimum pixels per grain are 2, which very often is not appropriate, especially if the authors want to define the grains based on the high angle grain boundaries. The reader does not know what did the authors used and he cannot repeat this study if he decided to do. If different grain definition is selected the grain size could be completely different. Please, clarify this point in the paper!
The sample coordinate system is added , but again this is incomplete. The readers are interested to know not what is x, y and z because they can be chosen arbitrary. The axes should be linked to sample processing i.e. compression axis -//z (probably) and the other two axes x and y . This should be clearly mentioned in the text or in the figure caption.
